# The development and pilot testing of an ACP simulation-based communication-training program: Feasibility and acceptability

Jui-O Chen[1,2], Shu-Chen Chang[3,4], Chiu-Chu Lin[5,6,7¤]*

**1** Department of Nursing, Tajen University, Pingtung County, Kaohsiung, Taiwan, **2** College of Nursing, Kaohsiung Medical University, Kaohsiung, Taiwan, **3** Department of Nursing, Changhua Christian Hospital, Changhua, Taiwan, **4** College of Nursing and Health Sciences, Dayeh University, Changhua, Taiwan, **5** School of Nursing, Kaohsiung Medical University, Kaohsiung, Taiwan, **6** Department of Renal Care, College of Medicine, Kaohsiung Medical University, Kaohsiung, Taiwan, **7** Department of Medical Research, Kaohsiung Medical University Hospital, Kaohsiung Medical University, Kaohsiung, Taiwan

¤ Current address: Sanmin Dist., Kaohsiung City, Taiwan
* chiuchu@kmu.edu.tw

**Data Availability Statement:** All relevant data are within the paper and its Supporting Information files.

## Abstract

The lack of knowledge of advance care planning and training of communication skills among nurses in Taiwan is one of the main reasons for the low rate of advance directive signing. However, there is no specific and effective solution to this problem. The purposes of this study were (1) to develop and pilot testing of an advance care planning simulation-based communication training program and (2) to evaluate the feasibility and acceptability of the program. This study was conducted in three phases. Phase 1: Developing an advance care planning simulation-based communication training program; Phase 2: Conducting a pilot test; Phase 3: Evaluating the feasibility and acceptability of the program. Twelve convenient participants from a medical center in central Taiwan were selected. The participants believed that team-based learning was beneficial for several reasons. First, it helped to clarify the participants' understanding of advance care planning and improve their communication skills. Second, role-playing, as one of the components, was helpful for discovering their own shortcomings in communication skills while debriefing enabled them to identify their blind spots in the communication process. Finally, the reflection log documented their weekly performance so they were able to reflect upon their weekly performance, improve their performance, and become more confident. All twelve participants signed the consent form and completed the whole training program. The participants were satisfied with the program, affirming that the timing and content of the program were appropriate and that the expected learning outcomes could be achieved. According to participant feedback, the program was beneficial in improving their knowledge of advance care planning and confidence in communication. Thus, it is feasible and acceptable to introduce communication of advance care planning programs into the staff training protocols of healthcare organizations.

**Clinical trial registration:** NCT04312295.

**Funding:** This study was funded by the Ministry of Science and Technology, Taiwan, R.O.C. (107-2314-B-037-029 -MY3). The funders had no role in study design, data collection and analysis, decision to publish, or preparation of the manuscript.

**Competing interests:** The authors have declared that no competing interests exist.

## Introduction

Chronic kidney disease (CKD) is a significant worldwide health issue with heavy economic burden [1]. In Taiwan, the incidence of end-stage renal disease (ESRD) is the highest in the world. ESRD is a difficult condition to cope with. For patients and their loved ones, end-of-life decisions may include patients' distressing inability to speak for themselves. Although there have been advances in hemodialysis technology, dialysis may not improve survival in elderly patients with multiple complications. Instead, it may affect quality of life [2]. Patients over 80 years of age with CKD have a mortality rate of up to 50% within one year of starting HD [3]. Ventilator-dependent CKD patients receiving HD have an in-hospital mortality rate of 69.6% and an extremely poor prognosis [4]. Some patients may also receive life-sustaining treatments, such as cardiopulmonary resuscitation and ventilation, which might not have been the patients' wishes.

The roles of advance directives (ADs) are two-fold. First, ADs help patients appoint a medical agent to speak on their behalf if they are unable to do so. Second, ADs work with patients and their families in clarifying and documenting the medical care they want towards the end of their lives [5]. The guiding principle for ADs is respect for autonomy. ADs are legal documents in which people can explicitly state what medical assistance they choose to receive or refuse. Despite the importance of ADs, it is estimated that advance care planning (ACP) internationally has only been implemented for 6% to 49% in CKD patients [6].

ACP is a dynamic communication process between patients, families, and healthcare professionals so that patients can receive the expected care when faced with uncertain treatments that they would or would not be willing to endure in the future [7]. Furthermore, ACP should be a patient-centered initiative that encourages shared decision-making, which may include patients completing an advance directive, documenting their wishes, and designating substitute decision makers [8].

The ESRD treatment process requires many complex decisions. Cognitive impairment is common in patients receiving long-term dialysis [9], so it is extremely important for patients to have a shared decision-making (SDM) relationship. The SDM can achieve the goals of fully informing the patient of the ethical treatment options, the possible risks, and benefits, and ensuring that the patient's values and preferences are taken into account in the medical decision-making process. Studies have demonstrated numerous benefits corresponding to the discussion of ACP, such as increased patient and family satisfaction with care [10] and the likelihood that physicians and families will understand and comply with patients' end-of-life care wishes [10, 11], thereby reducing "aggressive" medical care at the end of life [12].

However, there are considerable difficulties in promoting ACP and ADs in the clinical setting due to a variety of reasons. These include low public awareness of ACP, cultural taboo to talk about death [13], and lack of knowledge and communication skills among health care professionals [13, 14]. Although some studies have highlighted that the key factors hindering the discussion of ACP are the lack of knowledge of ACP among healthcare professionals and the inadequate and ineffective communication training, there are no specific and effective solutions, at this time.

To develop effective strategies for nurses to communicate ACP with patients, researchers utilized focus group qualitative interviews and literature reviews to integrate theories related to communication. Eneanya et al. noted that health care professionals in nephrology are limited in discussing end-of-life topics with patients due to lack of training on how to communicate prognosis with patients [15]. Therefore, in this study, an attempt was made to develop a theory-based communication training program to address this clinical dilemma, using cases of CKD patients as an example. The purposes of this study were (1) to develop and pilot testing

the ACP simulation-based communication training (ACP-SCT) program and (2) to assess the feasibility and acceptability of the program.

## Theoretical framework

Scaffolding theory is an instructional strategy used by experts to design short-term courses for specific learning content in which learners construct knowledge and acquire competencies through discussion and interaction with lecturers and peers, and through self-reflection, thus achieving the goal of building learners' self-constructed learning skills [16]. In addition, strategies such as cognitive construction, instruction, modeling, feedback, and questioning developed by scaffolding theory can be used for skill training [17].

Simulation has been widely used in the field of health education to develop the knowledge and skills of healthcare providers. It can be used individually in a program or in combination with other teaching strategies such as lectures, role plays, and interactive videos to enhance learning outcomes [18]. Studies have pointed out that to conduct a training program for ACP discussions, several aspects should be considered, including training in communication skills, group discussions, role plays, and the use of appropriate patient decision aids to enable practice of specific skills [19].

Bristowe et al. designed an advanced planning communication training course based on the PREPARED model and used it to guide the nephrology team in discussing end-of-life topics with patients with ESRD [20]. The PREPARED model is a communication guide based upon the letters of PREPARED used as a mnemonic to help remember the eight steps: **P**repare for the discussion, **R**elate to the person, **E**licit patient and caregiver preferences, **P**rovide information, **A**cknowledge emotions and concerns, **E**ncourage questions and further discussions, and **D**ocument [21]. Debriefing, though not part of the eight steps, is an indispensable part of the communication training process. After debriefing, learners reflect on their own communication process in four main areas: facts, feelings, findings, and future. For example, in which part did they do well? In which part did they find it difficult or uncertain? What can they do to improve in the future? The feedback loop is crucial as it facilitates learning by providing information to the learners and by realigning their thinking and behavior. It is also the key to promoting learning effectiveness and motivation [22].

Robinson et al. showed that the most effective way to conduct ACP is through one-on-one communication by trained professionals [23]. On the other hand, the best training method to familiarize health care workers with ACP is Team-Based Learning (TBL) [24]. The process of TBL implementation can be divided into three phases. First, the pre-class preparation phase is when the instructor gives the trainees the textbook and assigns preparation materials to review. Second, the reading assurance phase is when the instructor administers the Individual Reading Assurance Test (iRAT). iRAT is administered prior to the lesson so that the instructor knows the areas that might be more challenging for the trainees. After completing the iRAT, the Team Readiness Assurance Test (tRAT) is introduced to ensure the efficacy of *group* learning. At the end of the group test, the instructor circulates through the room and encourages groups to submit written appeals forms for the questions that they got wrong. The participants provide evidence from the study materials to support their alternate answers, thus they are afforded the opportunity to re-examine the study materials. This real-time feedback also allows the instructor to focus on the more challenging materials.

Third, the application phase is when teams of students solve, report, and discuss solutions to significant problems. By utilizing TBL's 4S problem-solving framework (significant problems, same problem, specific choice, and simultaneously report) to organize the problems, the instructor maximizes the power of team processing. Problems must be **significant** to ensure

the richness of discussion, while teams working on the **same problem** create opportunities for teams to challenge and learn from each other. **Specific choice** refers to the teams selecting the best choice from a limited list of options. **Simultaneous reporting** is accomplished with teams reporting their choice by holding up a colored card. Teams can then challenge each other while defending their own thinking.

The main purpose of TBL is to encourage students to apply their previously acquired knowledge in problem-solving. Through these team activities of solving, reporting, and discussing solutions to significant problems, members within the same group have opportunities for dialogue while teams have the opportunity to communicate with and learn from each other. Thus, the instructor can take on the role of clarifying and giving feedback in order to enhance student learning [25].

## Methods

In order to fulfill the research purpose, this study was conducted in three stages.

### Stage 1: Development of an ACP-SCT program

The ACP-SCT program had three main axes. The first axis was the conveying of professional knowledge, including the Hospice Palliative Care Act, the Patient Right to Autonomy Act, and the advance directive documents. The focus of the first axis was on enhancing the participants' knowledge related to palliative care for patients with renal disease, and commitment to the process. The second axis was the mastery of communication skills, including structured clinical communication through scenario simulation, role play, and guidance of the PREPARED model. The focus of the second axis was on improving participants' communication skills during the training. The third axis was debriefing and evaluation. Participants were encouraged to self-reflect on their communication skills while practicing ACP discussions with standardized patients and to self-correct based on feedback from peers and facilitators. In addition, participants kept a weekly reflective journal and recorded their self-perceptions, difficulties encountered, and insights gained from the feedback. The purpose of the weekly journal was to develop the participants' self-awareness and self-reflection skills.

**Expert validation.**    To determine the validity of the ACP-SCT program, expert validation was employed in this study. Three experts, a nephrologist, a hemodialysis unit head nurse, and a palliative care nurse, were invited to assess the validity of the program through a written review. The Content Validity Index (CVI) was used to evaluate the program design and content by considering the appropriateness and accuracy of content, semantic clarity, and clinical case scenario simulation. The scoring method was as follows: 1 = very inappropriate items that should be deleted, 2 = inappropriate items that should not be used in this case, 3 = acceptable items that require major modifications, 4 = appropriate items that require only minor modifications, 5 = very appropriate items that do not require modifications. This study utilized the CVI calculation proposed by Polit & Beck (2006) in which items scoring above 3 were retained and explanations and recommendations were added to items scoring below 3 [26]. The average CVI of this study was 0.89, which indicated that this study had excellent expert validity. The details of the ACP-SCT program are shown in Table 1.

### Stage 2: Pilot testing an ACP-SCT program

To prepare for future randomized controlled trials of the current study, we conducted a pilot study to test whether it could be done, and if so, how it should be done.

**Research design and participants.**    Twelve nurses, in two groups of six, were recruited through posters by convenience sampling from a hemodialysis unit at a medical center in

**Table 1. Content of the ACP-SCT program.**

| No. of Week | Content of Activities | Theoretical Basis |
|---|---|---|
| Week 1 (4 Hours) | **Goal: Improving the knowledge of learners** | • Scaffolding theory: Through course design, to equip participants with knowledge of ACP to discuss with CKD patients |
| | Activity 1. Individual Readiness Assurance Test (iRAT): The content of the test is as follows: | • TBL: To learn about the concepts of the course |
| | ✓ Advance care planning | |
| | ✓ Hospice Palliative Care Act/ Patient Right to Autonomy Act | ✓ iRAT: To assess the learner's initial understanding of the ACP curriculum |
| | ✓ Advanced directives and their completion instructions | ✓ tRAT: To develop the ability of group members to clarify issues, reach consensus, and ensure that learners know how to use the the knowledge they gained in the course preview |
| | ✓ Palliative care for patients with renal disease | |
| | Activity 2. Team Readiness Assurance Test (tRAT): The content of the tests is the same as above. | |
| | **Goal: Improving communication skills of learners** | • Scaffolding theory: Through course design, to equip participants with knowledge of ACP to discuss with CKD patients |
| | Activity 1. Practice of advanced communication skills and empathy | |
| | Activity 2. Introduction of the PREPARED model | |
| | Activity 3. Role play | • Use the PREPARED model as a guide for communication training |
| | Activity 4. Debriefing | • Role play: Participants practice communication skills in simulated scenarios |
| | Activity 5. Reflection | • Learners reflect on their strengths and weaknesses by reporting on their own performance during the communication process |
| | Activity 6. Discussion | |
| | • Homework: Reflective journal writing | • Based on feedback from other learners and the instructor, communication skills are realigned in order to consolidate learning |
| | | • Through discussion, participants ask clarifying questions about uncertain or confusing parts encountered |
| | | • Learners self-evaluate by recording their learning process, self-perceptions, difficulties encountered, and what they have learned from the feedback and results. |
| Week 2 (4 Hours) | **Goal: Knowing how to use the PREPARED model to guide the process of an ACP discussion** | • Under the guidance of the PREPARED model, learners practice continuously so that they can become proficient in communication skills |
| | Activity 1. Sharing reflective journal writing | • Peer learning is achieved through personal experience sharing and feedback among learners |
| | Activity 2. Role play | |
| | • Simulation scenario (I): Topics of refusing life-sustaining treatments | • By designing different simulation scenarios, learners can practice their communication skills with standardized patients |
| | • Simulation scenario (II): Topics of signing the document for ADs | |
| | Activity 3. Debriefing | • Same theoretical basis as Week 1 |
| | Activity 4. Reflection | |
| | Activity 5. Discussion | |
| | • Homework: Reflective journal writing | |
| Week 3 (4 Hours) | **Goal: Becoming proficient in applying the PREPARED model** | • Through repeated practice, learners become fluent in ACP discussions using the PREPARED model |
| | Activity 1. Sharing reflective journal writing | |
| | Activity 2. Role play | |
| | • Simulation scenario (III): Topics of weaning life-sustaining treatment | • Same theoretical basis as Week 2 |
| | • Simulation scenario (IV): Topics of palliative and home care | |
| | Activity 3. Debriefing | |
| | Activity 4. Reflection | |
| | Activity 5. Discussion | |

central Taiwan. All the participants received a three-week ACP-SCT course, which was conducted once a week for four hours.

**Group leaders.** The ACP-SCT program workshop was conducted by a facilitator, the first author, specializing in palliative care and an ACP counselor. Another researcher, the corresponding author, assumed the role of an observer and recorded the whole process of the

workshop. After the pilot study, the research team made changes to the program and the facilitators' strategies after a thorough group discussion.

**Program content and process.** This study was reviewed by the Medical Center Ethics Committee (IRB No. 190115). The study was initiated after the researchers obtained signed informed consent from the participants. The ACP-SCT program workshops were conducted over three consecutive weeks, with one 4-hour session per week. During the first week, the sessions were conducted in a TBL format with the goal of improving ACP knowledge and communication skills. During the second and third weeks, the sessions were conducted in breakout groups with the goal of becoming more proficient in the application of communication skills and conducting ACP discussions with standardized patients. Prior to the start of the role-play activity for the week, participants were asked to share their reflections from the weekly journal. After sharing, feedback was given by the facilitator and other participants in the group. Next, each participant took turns applying the PREPARED model and role-played with standardized patients based on various scenarios. Afterwards, the participants reflected on their perceptions, strengths, and weaknesses during the interaction with the standardized patient. Feedback was then given by the facilitator and other members in the group. Participants took approximately 25 to 30 minutes to complete their turn and the session ended when all six participants in the group completed their role-play exercise.

## Stage 3: Evaluation of the feasibility and acceptability of the implementation of an ACP-SCT program

We assessed the perceived feasibility and acceptability of the intervention activities and program implementation through a workshop with hemodialysis nurses.

**Feasibility.** The study by Gilissen et al. suggested that program feasibility can be measured based on the progress of the intervention in the original plan [27]. The feasibility measures of the ACP-SCT program in this study were informed by Khumsaen & Stephenson's study and included: (a) participant recruitment rate, (b) percentage of interested participants willing to provide consent, and (c) participant retention rate [28].

**Acceptability.** Acceptability is a multidimensionally structured index that reflects participants' expectations, experiential perceptions, and affective responses to an intervention. Sidani et al. suggested that several factors influence participants' perceptions of the acceptability of an intervention prior to their participation. Factors such as participants' attitudes toward the intervention, the appropriateness, suitability, and perceived effectiveness of the intervention were considered as indicators of treatment acceptability [29].

Researchers used three open-ended questions to measure the acceptability of the program. After receiving the three-week workshop, participants were asked to answer the following questions: (1) Describe changes or breakthroughs in their communication skills; (2) Identify the activities in the workshop that improved their communication abilities and confidence; and (3) Describe the benefits they gained from the ACP-SCT program.

## Results

### Demographics

A total of 12 females participated with an average age of 44.6 years (range = 26–55 years). 91.7% (n = 11) of them had received a university education; 83.3% (n = 10) were married; and 66.7% (n = 8) held religious beliefs. In terms of clinical experience, the average was 22.2 years (range = 4–35 years). Though 83.3% (n = 10) of the participants had received palliative care training, 67.7% (n = 8) had never received courses about breaking bad news or the Patient Right to Autonomy Act.

## Feasibility

After the participants completed the three-week workshop, their participation rates were calculated. The results were as follows: (a) 100% participant recruitment rate, (b) 100% of interested participants were willing to provide consent, and (c) there was a 100% retention rate of participants who attended all sessions.

## Acceptability

All participants answered three open-ended questions, and the results of the qualitative analysis are presented below.

**Personal changes or breakthroughs in communication skills.** After receiving the training from the ACP-SCT program, participants conscientiously reported changes in their communication skills. These changes included: (a) *learning how to patiently listen to patients' ideas.* As participant #1 noted, the impulse to rush through the work has eased and instead more attention has been shifted to patiently listening to patients' ideas; (b) *learning how to infer the connotations of patients' messages.* Three participants noted that previously, when talking with patients, they felt they could only grasp the literal meaning. For example, participant #2 commented, "Now, I can catch the key words in the patient's messages and encourage them to expand on their thinking." Participant #3 concurred, "I used to respond out of instinct. Now, I am more able to reflect the true meaning behind the patient's message." In addition, participant #5 described, "The biggest change for me is that I will now always try to figure out what the patient is really trying to say through the conversation." (c) *Learning how to use communication skills in the appropriate way at the appropriate time.* After receiving self-awareness training, many participants' communication skills improved significantly. For example, they were more able to use conversational skills when needed, including asking clarifying questions, paying closer attention, and being more empathetic. As an example, Participant #6 mentioned, "I'm a more empathetic listener than before. In the past, when I encountered problems that I had no solution to, I would choose to run away. Now, I can put myself in another person's shoes and be in tune with them." Participant #8 also explained, "Now, I can compliment my patients when they change in favor of meeting expressed goals. In addition, I am more willing to engage in self-disclosure, share successful experiences, and provide the appropriate information to patients"; (d) *Learning how to discern and respond to patients' emotions.* Participant #9 noted, "Previously, when my patients or their families were in a negative mood, I had no recourse. Now I can recognize what kind of emotions my patients are experiencing and respond accordingly." Participant #4 also claimed, "I feel more encouraged to explore the situations of patients who are depressed or facing imminent death. I am also more comfortable talking to patients about end-of-life topics."

According to the responses of the study participants, the ACP-SCT program did significantly improve their communication skills. They can now use these skills when needed and are able to make a breakthrough from avoidance to a welcoming attitude when encountering problems, negative emotions, or end-of-life topics brought up by patients.

**Activities that improved their communication abilities and confidence.** The activities designed in the ACP-SCT program included the elements of TBL, role-play, reflection, and feedback. The following activities improved the participants' communication abilities and confidence.

*TBL.* Through inter- and intra-group sharing, Participant #12 mentioned, "Guided learning through two-way communication was helpful in clarifying my understanding of ACP and improving my communication skills." Participant #10 also noted, "Learning more multidimensional communication skills through TBL improved my abilities to start dialogues with my

patients." Indeed, Participant #1 offered, "The information in the handbook has improved my knowledge and communication skills."

*Role play*. Through the process of role play with standardized patients under scenario simulation, participants were given the opportunity to develop critical thinking and learn from the cases. Participant #2 stated, "The role-play was very well designed in which I could recognize my blind spots through self-reflection and feedback from the instructor." In addition, role play provides a sense of authenticity to the participants. Participant #12 mentioned, "The role play activities reflect those situations that I would encounter in a real clinical setting where I can practice to improve my communication skills." Moreover, role play provided opportunities for peer observation and learning. Participant #7 noted, "During the role play, I was able to observe other participants which gave me a better understanding of issues that I could not or barely notice when interacting with patients."

*Reflection*. Through the process of self-reflection and recording the results, participants can recognize their strengths and weaknesses. As Participant #11 observed, "I used to think that self-reflection was a lot of work. Now I realize that by reviewing my performance in the previous week, I can better understand my shortcomings and learn from others, thus building my confidence." Participant #6 concurred, "By reflecting on my work, I can know what I can do better next time, which is helpful to my patients and their families."

*Feedback*. Feedback is the core of scenario training because it helps participants have a clear understanding of the problem, thus enabling correcting and enhancing of the learning effect. For example, Participant #12 explained, "I am more aware of my abilities and what to do in the future when I encounter similar problems in an authentic clinical setting." Participant #5 also mentioned, "We have a better understanding of the key points and know the appropriate time to intervene with patients and their families in ACP." Participant #7 offered, "The feedback I received from the instructor was valuable in guiding me to know how to identify emotions both mine and the patient's, cope with them, and become more confident in the end."

**Benefits gained from the ACP-SCT program.** Participants' scores indicated the **facilitative** effect of the ACP-SCT program on their discussions of ACP with patients in authentic clinical settings. When asked to score on a scale of 0–10 (0 being the lowest and 10 as the highest), participants' scores ranged from 5 to 9, with a mean score of 7.8.

## Discussion

The ACP-SCT program built scenarios that simulated authentic clinical situations through simulations, allowing participants to interact with standardized patients without worrying about hurting patients' feelings if participants interacted inappropriately. In addition, the instructor provided immediate feedback to participants for realignment, allowing them to learn how to communicate effectively in a controlled environment. This finding was in line with Kirkham's research, in which he claimed that training provided through scenario simulation had many benefits. These benefits included allowing trainees to practice skills in a regulated environment, becoming aware of the consequences of their actions, developing their cognitive skills, building their decision-making processes, ensuring effective communication, and self-correcting based on immediate feedback from the instructor [30].

The participants felt that the program had a positive impact on them in many ways and would be beneficial to their future discussions of ACP with patients in the clinical setting. The TBL approach elicited interaction between the participants and the instructor, which led to active learning and laid the foundation for ACP knowledge and communication skills. The role play reflected an authentic clinical setting. By practicing one-on-one with standardized patients in different scenarios, participants not only recognized their own strengths and

weaknesses in the communication process, but also observed and learned from other participants' role play experiences. The findings of TBL and role play confirmed what Decker et al. observed in a simulated, interactive learning environment in that it could reinforce active motivation and confidence in learning [31]. It also corresponded with Neill & Wotton's view that scenario-based simulation training provided a link between theory and real-life environments and facilitated the development of clinical decision-making skills by student nurses [32].

Debriefing is an essential component of learning and provides learners with the opportunity to engage in self-reflection and translate learning experiences into practical knowledge [31]. Training using scenario simulation encourages learners to reflect on their performance and allows the participants to think about the validity of their decisions [33]. Debriefing can also improve self-awareness and problem-solving skills, thereby developing critical thinking skills [34].

In this study, the 4F's of facts, feelings, findings, and future were used as the focus of reflection, and participants were asked to self-reflect on their performance after the activity. Participants were also asked to write weekly reflective essays to analyze their perceptions and difficulties encountered in clinical practice, and then to evaluate their learning outcomes. Participants noted that by maintaining weekly reflective writing, they could reflect on their performance, enhance their self-awareness, realign themselves based on feedback from others, and become more confident. This finding coincided with Craft's theory that reflective journaling contributed to the development of critical thinking skills and the acquisition of professional experience [35].

Feedback is central to scenario-based training as it directs participants' attention to blind spots in their performance, helps them to know how to solve similar problems in the future, and enhances learning outcomes. Based on participants' feedback on the program content design, this ACP-SCT program's theoretical basis was validated with reasonable feasibility and acceptability.

## Limitations

ACP is a communication process in which various health professionals, such as physicians, nurse practitioners, social workers, and psychologists, discuss with patients and their families their personal preferences for future medical care in the face of loss of decision-making capacity. Accordingly, ACP is a person-centered, interdisciplinary, and collaborative approach to patient care that involves professionals of diverse backgrounds and areas of expertise. However, due to time, personnel, and resource constraints, only the nurses who are often the frontline caregivers were prioritized for training to test the feasibility and acceptability of this ACP-SCT program. It is recommended that future research could develop an ACP training program derived from our protocol but further includes a variety of healthcare professionals to validate the team's central role in ACP.

## Conclusions

In this study, the researchers conducted a pilot study in preparation for a future definitive randomized clinical trial. The results demonstrated that participants affirmed that the workshop was beneficial for improving ACP-related knowledge and communication confidence and skill. The results further confirmed that this ACP-SCT program would be feasible and acceptable for future staff training regarding ACP discussions in healthcare organizations.

## Supporting information

**S1 File.**
(PDF)

## Acknowledgments

The authors are grateful to all participants who shared their valuable experiences in this study.

## Author Contributions

**Conceptualization:** Chiu-Chu Lin.

**Investigation:** Jui-O Chen, Shu-Chen Chang.

**Methodology:** Jui-O Chen, Chiu-Chu Lin.

**Project administration:** Chiu-Chu Lin.

**Resources:** Shu-Chen Chang.

**Supervision:** Chiu-Chu Lin.

**Writing – original draft:** Jui-O Chen.

**Writing – review & editing:** Shu-Chen Chang, Chiu-Chu Lin.

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
