## [Decision Letter · Decision Letter 0]

9 Feb 2021

PONE-D-20-39392

The development and pilot testing of an ACP simulation-based communication-training program: feasibility and acceptability

PLOS ONE

Dear Dr. Chiu-Chu Lin,

Thank you for submitting your manuscript to PLOS ONE. After careful consideration, we feel that it has merit but does not fully meet PLOS ONE’s publication criteria as it currently stands. Therefore, we invite you to submit a revised version of the manuscript that addresses the points raised during the review process.

We look forward to receiving your revised manuscript.

Kind regards,

Alessandra Solari, M.D.

Academic Editor

PLOS ONE

Journal Requirements:

2.Thank you for including your ethics statement: 

"This study is reviewed by the Ethics Committee of Changhua Christian Hospital (IRB No. 190115)".   

Please amend your current ethics statement to confirm that your named institutional review board or ethics committee specifically approved this study. Once you have amended this/these statement(s) in the Methods section of the manuscript, please add the same text to the “Ethics Statement” field of the submission form (via “Edit Submission”).

3. Thank you for stating in the text of your manuscript "The study was preceded after the researchers obtained informed consents from the study participants and agreements were signed." Please also add this information to your ethics statement in the online submission form.

4. Please provide training and educational materials used for the three phases of the study.

5. We suggest you thoroughly copyedit your manuscript for language usage, spelling, and grammar. If you do not know anyone who can help you do this, you may wish to consider employing a professional scientific editing service.  

The name of the colleague or the details of the professional service that edited your manuscriptA copy of your manuscript showing your changes by either highlighting them or using track changes (uploaded as a *supporting information* file)

Reviewers' comments:

Reviewer's Responses to Questions

**Comments to the Author**

1. Is the manuscript technically sound, and do the data support the conclusions?

Reviewer #1: Yes

Reviewer #2: Partly

Reviewer #3: Partly

2. Has the statistical analysis been performed appropriately and rigorously? 

Reviewer #1: Yes

Reviewer #2: Yes

Reviewer #3: N/A

3. Have the authors made all data underlying the findings in their manuscript fully available?

Reviewer #1: Yes

Reviewer #2: Yes

Reviewer #3: Yes

4. Is the manuscript presented in an intelligible fashion and written in standard English?

Reviewer #1: Yes

Reviewer #2: Yes

Reviewer #3: No

5. Review Comments to the Author

Reviewer #1: The article describes a very interesting project.

The analysis reports that the program can be used as training material for communication of ACP in the future, to promote ACP and advanced directives in clinical settings.

It would be interesting to extend the study to a broader change, and to verify after some time whether there is an increase to ACP in care settings

Reviewer #2: The research is carried out with attention at each learning step and effective for the purpose of changing attitudes in the relationship with the patient in communicating the bad news. I think the method is very interesting, even if it is very demanding of resources. I wonder how much it is possible to replicate it in other formative realities.

I would expand and argue the conclusion more.

Reviewer #3: With the limitations of my English and my scarce knowledge of some part of the theoretical framework and of methods to calculate content validity index, I had some problems in getting the information contained in the text.

See attached comments.

6. PLOS authors have the option to publish the peer review history of their article (what does this mean?). If published, this will include your full peer review and any attached files.

Reviewer #1: **Yes: **Marta Cascioli

Reviewer #2: **Yes: **Barbara Lissoni

Reviewer #3: **Yes: **eugenio pucci

---

## [Author Response · Author response to Decision Letter 0]

10 Apr 2021

Reviewers’ Comments and Response to Reviewers

I have incorporated all of your comments and suggestions into my revision. There were very helpful and improve the quality of the article. Thank you very much.

•24 ESRD: it is better to explain the abbreviation the first time it appears in your article. 

Not only death …It could be better to explain that ESRD is a condition difficult to cope with, and distressing for the patients and their significant ones, with end-of –life decisions which can include situation in which patients are not able to speak for themselves, and so on…

p.4 (Lines 26-29) 

Revised as follows:

Chronic kidney disease (CKD) is a significant worldwide health issue with heavy economic burden [1].In Taiwan, the incidence of end-stage renal disease (ESRD) is the highest in the world. ESRD is a difficult condition to cope with. For patients and their loved ones, end-of-life decisions may include patients’ distressing inability to speak for themselves. 

29. “If advance care planning (ACP) can be discussed with patients and the ADs can be signed, patients may avoid the suffering of medical futility”

This statement is incorrect. ACP does not concern with medical futility.

p.4 (Lines 35-40)

Revised as follows: 

The roles of advance directives (ADs) are two-fold. First, ADs help patients appoint a medical agent to speak on their behalf if they are unable to do so. Second, ADs work with patients and their families in clarifying and documenting the medical care they want towards the end of their lives [4]. The guiding principle for ADs is the respect for autonomy. ADs are legal documents in which people can explicitly state what medical assistance they choose to receive or refuse. Despite the importance of ADs, it is estimated that advance care planning (ACP) internationally has only been implemented for 6% to 49% in CKD patients [6].

31 I suggest to delete the term “medical” to consider care in an holistic approach aligned with patients’ preferences and values. It should be stressed tat ACP differs from general medical decision-making in being based on the person’s wishes for the time when their decisional capacity is impaired. As far as the family is concerned, it should be better to say that the family can be involved in the process, if the patient wishes... I understand that different cultural, ethical and legal issues may be important in how the family is involved in ACP.

p.4 (Line 42) 

We agreed with the reviewer's suggestion and deleted the term “medical”

32 disability 

I suggest to use terms such as “severe impairment” and/or “new condition of life” and/or “treatments they would or would not be willing to endure”.

p.4 (Line 42-43) 

We agreed with the reviewer's opinion. 

Revised as follows:

ACP is a dynamic communication process between patients, families, and healthcare professionals so that patients can receive the expected care when faced with an uncertain treatments that they would or would not be willing to endure in the future.

33 The phrase sounds wrong and should be reformulated to underline the prominent ethical value of ADC?, that is autonomy.

p.5 (Lines 46-53) 

Revised as follows:

The ESRD treatment process requires many complex decisions. Cognitive impairment is common in patients receiving long-term dialysis [9], so it is extremely important for patients to have shared decision-making (SDM) relationship. The SDM can achieve the goals of fully informing the patient of the ethical treatment options, the possible risks, and benefits, and ensuring that the patient's values and preferences are taken into account in the medical decision-making process. Studies have demonstrated numerous benefits corresponding to the discussion of ACP, such as increased patient and family satisfaction with care [10] and the likelihood that physicians and families will understand and comply with patients' end-of-life care wishes [10,11], thereby reducing "aggressive" medical care at the end of life [12].

44 I suggest to remove "including the scaffolding theory, the scenario simulation model, and the PREPARED model, as the basis of the ACP simulation-based communication training (ACP-SCT) program and adopted team-based learning (TBL) as the teaching method for program training”. It is confusing to me to have such terms anticipated and I can read about them easily in the next section “Theoretical framework “

p.5 (Line 62) 

Removed "including the scaffolding theory, the scenario simulation model, and the PREPARED model, as the basis of the ACP simulation-based communication training (ACP-SCT) program and adopted team-based learning (TBL) as the teaching method for program training”.

70 Robinson et al 

p.6 (Line 92) 

Added “et al”

73-76 “ask them to preview the assigned reading materials before classes. In the second phase, learners will be asked to undertake the individual reading assurance test before classes so that lecturers can understand the learning outcomes before classes. After the individual test, the group readiness assurance test will be introduced to ensure the efficacy of group learning and to obtain the effect of real-time feedback. ask them to preview the assigned reading materials before classes “ 

I don’t understand… Please help me … Revised the section: p.6 (Line 94) to p.7 (Line 111)

The process of TBL..... Teams can then challenge each other while defending their own thinking.

81 Eneanya et al

p.5 (Line 62 )

Added “et al”

81-86 I suggest to move at the end of the introduction 

p.5 (Line 62) to p.6 (Line 67) 

Moved the following to the end of the introduction. 

“Eneanya et al. noted that health care professionals in nephrology are limited in discussing end-of-life topics with patients due to lack of training on how to communicate prognoses with patients [15]. Therefore, in this study, an attempt was made to develop a theory-based communication training program to address this clinical dilemma, using cases of CKD patients as an example. The purposes of this study were (1) to develop and pilot the ACP simulation-based communication training (ACP-SCT) program and (2) to assess the feasibility and acceptability of the program.”

112 Polit (typo)

p.8 (Line 141) 

Fixed typo, changed ‘polit’ to ‘pilot’ 

150 The way in which recruitment was done is not clear. Probably as a consequence of this, I don’t understand the difference between recruitment and consent to participate. Moreover since the 12 nurses were enrolled by convenience sampling it is difficult to think that recruitment may be a way to measure feasibility

For this study, a convenience sample was taken and the researcher visited the head of the nephrology unit of a medical center to explain the purpose of the study. Posters were used to promote the study to the nursing staff. QR code was included to facilitate the enrollment of interested nurses. The number of recruits was reached within one week.

The feasibility measures of the ACP-SCT program in this study followed Khumsaen & Stephenson's study and included (a) recruitment rate of participants, (b) percentage of interested participants willing to provide consent, and (c) retention rate of participants. 

283 It should be correct to consider this paper as a first step within a research program. Is available the protocol of the “future definitive randomized clinical trial”? Is that trial protocol registered in a trail registry?

The current study protocol has been registered on: Clinical Trails.gov PRS.

Clinical Trails.gov ID: NCT 04312295

I think that educational program in palliative care and in ACP should involve all the health professionals at the same time (in this case both physician and nurse from hemodialysis room). Training programs could be more effective if carried out in multi-professional equipe dealing with with communication problems among heath professionals. In fact, a challenge in the ACP process may be the different approach of the health professionals involved in patients’ care for several reasons (different values, different skill, hierarchy, team flexibility and so on).(1 ) Was this issue considered in the goal “improve communication skills of learners”? Did the ACP-STC program deal with team communication and factors within the health professional team that influence the success of the ACP process? How much concern was given to the item “D” of the PREPARED recommendations?

Q: Was this issue considered in the goal “improve communication skills of learners”? Did the ACP-STC program deal with team communication and factors within the health professional team that influence the success of the ACP process?

A: Although it is the responsibility of all healthcare professionals to promote ACP, the researcher found through literature review and focus group interviews that nursing staff are the frontline healthcare providers. They spend the most time with patients and families and are the most trusted. However, the nursing staff are nervous to discuss ACP with patients and families due to lack of communication skills. Given manpower and resource constraints, the researcher started with the training of nursing staff. If the ACP- SCT program is effective, it can then be widely applied to other healthcare professionals.

Q: How much concern was given to the item “D” of the PREPARED recommendations?

A: 1. Write a summary of what was discussed in the medical record.

2. Talk or write to other key healthcare providers involved in the patient’s care. 

3. The IC health insurance card will be used to record a patient’s medical wishes so that health care providers do not go against the patient's wishes.

•Please amend your current ethics statement to confirm that your named institutional review board or ethics committee specifically approved this study. Once you have amended this/these statement(s) in the Methods section of the manuscript, please add the same text to the “Ethics Statement” field of the submission form (via “Edit Submission”).

•Thank you for stating in the text of your manuscript "The study was preceded after the researchers obtained informed consents from the study participants and agreements were signed." Please also add this information to your ethics statement in the online submission form. 

The IRB statement and informed consent form have been amended in the Methods section and submitted on the "Ethics Statement".

Reviewer #3: With the limitations of my English and my scarce knowledge of some part of the theoretical framework and of methods to calculate content validity index, I had some problems in getting the information contained in the text. In this study, the Content Validity Index (CVI) was used to evaluate the program design and content by considering the appropriateness and accuracy of content, semantic clarity. 

CVI calculation= The number of experts which item scoring above 3/ total number of experts.

Appropriateness of content=3/3

Accuracy of content=2/3

Semantic clarity=3/3

CVI=(1+0.67+1)/3=0.89

Please provide training and educational materials used for the three phases of the study. 

The handbook of the ACP-SCT program totals 63 pages. Since the target subjects of this study are Taiwanese, the training and teaching materials for this study are in Chinese. If you are interested in any part, we will try our best to provide the English version you need.

We suggest you thoroughly copyedit your manuscript for language usage, spelling, and grammar. 

The manuscript has been carefully reviewed by experienced editors who specialize in editing papers written by researchers whose first language is not English.

---

## [Decision Letter · Decision Letter 1]

15 Jun 2021

PONE-D-20-39392R1

The development and pilot testing of an ACP simulation-based communication-training program: feasibility and acceptability

PLOS ONE

Dear Dr. Lin,

Thank you for submitting your manuscript to PLOS ONE. After careful consideration, we feel that it has merit but does not fully meet PLOS ONE’s publication criteria as it currently stands. Therefore, we invite you to submit a revised version of the manuscript that addresses the points raised during the review process.

We look forward to receiving your revised manuscript.

Kind regards,

Gwo-Jen Hwang

Academic Editor

PLOS ONE

Journal Requirements:

Reviewers' comments:

Reviewer's Responses to Questions

**Comments to the Author**

1. If the authors have adequately addressed your comments raised in a previous round of review and you feel that this manuscript is now acceptable for publication, you may indicate that here to bypass the “Comments to the Author” section, enter your conflict of interest statement in the “Confidential to Editor” section, and submit your "Accept" recommendation.

Reviewer #1: All comments have been addressed

Reviewer #3: (No Response)

2. Is the manuscript technically sound, and do the data support the conclusions?

Reviewer #1: Yes

Reviewer #3: Yes

3. Has the statistical analysis been performed appropriately and rigorously? 

Reviewer #1: Yes

Reviewer #3: N/A

4. Have the authors made all data underlying the findings in their manuscript fully available?

Reviewer #1: Yes

Reviewer #3: Yes

5. Is the manuscript presented in an intelligible fashion and written in standard English?

Reviewer #1: Yes

Reviewer #3: Yes

6. Review Comments to the Author

Reviewer #1: It would be important in the future to carry out a training and simulation path with different professionals with different training and backgrounds, to validate the central role of the team in advanced care planning.

Reviewer #3: Sorry, but I insist on the concept that educational program in ACP should involve all the health professionals at the same time who are involved in the patients' care. I would like that the Authors discuss this issue/limit of the their study in the discussion.

---

## [Author Response · Author response to Decision Letter 1]

7 Jul 2021

Reviewer’s comments Response to reviewers

 Reviewer #1: It would be important in the future to carry out a training and simulation path with different professionals with different training and backgrounds, to validate the central role of the team in advanced care planning.

 Agreed. Added the reviewer’s comments to the limitations paragraph. (Page 15, lines 308-317)

Reviewer #3: Sorry, but I insist on the concept that educational program in ACP should involve all the health professionals at the same time who are involved in the patients' care. I would like that the Authors discuss this issue/limit of their study in the discussion.

 Agreed. Thank you for the very important and specific suggestion. The limitations paragraph has been added to this study: (Page 15, lines 308-317)

ACP is a communication process in which various health professionals, such as physicians, nurse practitioners, social workers, and psychologists, discuss with patients and their families their personal preferences for future medical care in the face of loss of decision-making capacity. Accordingly, ACP is a person-centered, interdisciplinary, and collaborative approach to patient care that involves professionals of diverse backgrounds and areas of expertise. However, due to time, personnel, and resource constraints, only the nurses who are often the front-line caregivers were prioritized for training to test the feasibility and acceptability of this ACP-SCT program. It is recommended that future research could develop an ACP training program derived from our protocol but further includes a variety of healthcare professionals to validate the team's central role in ACP.

---

## [Editor Report · Decision Letter 2]

8 Jul 2021

The development and pilot testing of an ACP simulation-based communication-training program: feasibility and acceptability

PONE-D-20-39392R2

Dear Dr. Lin,

We’re pleased to inform you that your manuscript has been judged scientifically suitable for publication and will be formally accepted for publication once it meets all outstanding technical requirements.

Kind regards,

Gwo-Jen Hwang

Academic Editor

PLOS ONE
---

## [Editor Report · Acceptance letter]

13 Aug 2021

PONE-D-20-39392R2 

The development and pilot testing of an ACP simulation-based communication-training program: feasibility and acceptability 

Dear Dr. Lin:

I'm pleased to inform you that your manuscript has been deemed suitable for publication in PLOS ONE. Congratulations! Your manuscript is now with our production department. 

Kind regards, 

on behalf of

Dr. Gwo-Jen Hwang 

Academic Editor

PLOS ONE